# Controllable deuteration of halogenated compounds by photocatalytic D$_2$O splitting

Cuibo Liu[1,2], Zhongxin Chen[2,3], Chenliang Su [1,2], Xiaoxu Zhao [2,3], Qiang Gao [1,2], Guo-Hong Ning[2], Hai Zhu[2], Wei Tang[4], Kai Leng[2], Wei Fu[2], Bingbing Tian [1,2], Xinwen Peng[2], Jing Li [1,2], Qing-Hua Xu[1,2], Wu Zhou[5] & Kian Ping Loh[1,2]

Deuterium labeling is of great value in organic synthesis and the pharmaceutical industry. However, the state-of-the-art C–H/C–D exchange using noble metal catalysts or strong bases/acids suffers from poor functional group tolerances, poor selectivity and lack of scope for generating molecular complexity. Herein, we demonstrate the deuteration of halides using heavy water as the deuteration reagent and porous CdSe nanosheets as the catalyst. The deuteration mechanism involves the generation of highly active carbon and deuterium radicals via photoinduced electron transfer from CdSe to the substrates, followed by tandem radicals coupling process, which is mechanistically distinct from the traditional methods involving deuterium cations or anions. Our deuteration strategy shows better selectivity and functional group tolerances than current C–H/C–D exchange methods. Extending the synthetic scope, deuterated boronic acids, halides, alkynes, and aldehydes can be used as synthons in Suzuki coupling, Click reaction, C–H bond insertion reaction etc. for the synthesis of complex deuterated molecules.

[1] SZU-NUS Collaborative Center and International Collaborative Laboratory of 2D Materials for Optoelectronic Science & Technology, Engineering Technology Research Center for 2D Materials Information Functional Devices and Systems of Guangdong Province, College of Optoelectronic Engineering, Shenzhen University, Shen Zhen 518060, China. [2] Department of Chemistry and Centre for Advanced 2D Materials (CA2DM), National University of Singapore, 3 Science Drive 3, Singapore 117543, Singapore. [3] NUS Graduate School for Integrative Sciences and Engineering, National University of Singapore, Centre for Life Sciences, #05-01, 28 Medical Drive, Singapore 117456, Singapore. [4] Institute of Materials Research and Engineering, 2 Fusionopolis Way, Singapore 138634, Singapore. [5] School of Physical Sciences, CAS Key Laboratory of Vacuum Physics, University of Chinese Academy of Sciences, Beijing 100049, China. Cuibo Liu and Zhongxin Chen contributed equally to this work. Correspondence and requests for materials should be addressed to C.S. (email: chmsuc@szu.edu.cn) or to K.P.L. (email: chmlohkp@nus.edu.sg)

Owing to the kinetic isotope effect, deuterium- and tritium-labeled compounds find important applications in the investigation of reaction mechanism[1], the modification of reaction selectivity in total synthesis[2], as well as the preparation of advanced materials with enhanced performance[3]. In particular, D/T labeling has attracted growing interests in pharmaceutical industry because the incorporation of heavy hydrogen can change the absorption, distribution, and toxicological properties of the drugs, while retaining their original potency and selectivity owing to the longer residence time of the more inert C–D bonds compared with the C–H isotopologues[4–6]. Since the first deuteration of bioactive molecules to promote improved pharmacokinetic profile in the 1960s[7], tremendous efforts have been devoted to the synthesis of deuterium- or tritium-labeled pharmaceuticals. Owing to the prevalence of aryl moiety in drug molecules[8], direct deuteration of aryl ring would be promising for generating new drugs with high therapeutic values. Indeed, the first deuterated drug has been approved by the US Food and Drug Administration in April 2017[9].

The prevalent methods for deuterium incorporation are metal-catalyzed or acid/base promoted C–H/C–D exchanges[10], which allow direct incorporation of deuterium without the need to pre-functionalize the starting materials and without substantially altering the structure of the molecules. Nevertheless, conventional C–H/C–D exchanges suffer from several drawbacks. First, C–H bonds are usually difficult to activate. Industrial C–H/C–D exchanges relying on the weak acidity or basicity of C–H bonds require the use of high temperature and strong acid/base, leading to significant safety risks and poor functional group tolerances[11,12]. Multiple exchange processes are also necessary to achieve high deuteration content[13]. Second, approaches using homogeneous noble metal catalysts to activate C–H bonds are restricted by the complexity of catalyst synthesis and high cost[14,15]. Third, the selective deuteration of C–H bonds remains challenging; the position of deuteration is difficult to control and requires directing groups to install deuterium at the ortho positions on aromatic and hetero-aromatic rings[16]. Therefore, undesired deuteration at multiple reactive sites is difficult to prevent[17,18]. The development of a universal deuteration strategy with wide functional group tolerance will be highly valuable in synthetic chemistry and pharmaceutical industry.

C–X (where X is a halogen) bonds are ubiquitous in many important organic molecules and natural products[19]. In fact, well-developed halogen chemistry can be exploited for selective deuteration[20]. For example, Onomura group developed a Pd-NHC system for effective deuterodechlorination of aryl/heteroaryl

chlorides[21]. However, C–X to C–D transformations generally require the use of noble metal catalysts, complex ligands and special deuterium donors. Recently, photoinduced electron transfer from excited photocatalysts to halides (C–I, C–Br, C–Cl, and C–F) has emerged as a powerful strategy to generate highly reactive carbon radicals, which are useful intermediates for noble metal-free substitution, addition, or coupling reactions to produce value-added compounds[22]. A comprehensive review on the photoredox reactions used in organic and polymer synthesis is provided by Xu and Boyer et al[23]. Owing to their mild reaction conditions and high efficiency, photocatalytic methods have been successfully applied in total synthesis or drug discovery process[24,25]. Despite some early studies on photocatalytic hydrogenation of Ar–X using amines, alcohols, or Hantzsch ester as hydrogen donors[26–28], photochemical deuteration has not been widely studied owing to the difficulty in accessing suitable deuterium sources and the lack of a suitable catalytic system. Heavy water ($D_2O$) is an ideal deuteration reagent from the perspective of safety, cost and handling and it is worthwhile exploring photocatalytic hydrogen evolution (PHE) as a potential route for generating active D radicals for deuteration reactions.

Herein, we report a universal C–X to C–D transformation using $D_2O$ as deuterium source and porous 2D-CdSe as photocatalyst; the latter can catalyze C–X bonds cleavage as well as split $D_2O$ to generate deuterium radicals. The subsequent coupling between carbon and deuterium radicals results in the formation of deuterated products. Compared with conventional C–H/C–D exchange technology, this method is distinguished by its high selectivity, good functional group tolerance, and ability to undergo intramolecular tandem reaction in mild conditions.

## Results

**Porous CdSe nanosheet photocatalyst.** The C–X to C–D transformation proposed here relies on the effective photocatalytic splitting of $D_2O$ to generate active D radicals (Fig. 1 and Supplementary Table 1). CdSe is selected for its appropriate bandgap for solar energy absorption and a conduction band edge that is more negative in potential than water reduction potential[29,30]. Reducing its dimension from bulk to nanoscale further improves its catalytic performance owing to quantum confinement effect. Current synthetic methods for CdSe nanosheets utilize organic ligands to control the shape and size of the nanocrystals. However, such ligands invariably block or deactivate the active catalytic sites. To increase the density of catalytic sites, porosity can be introduced into 2D-CdSe, as the edges of these pores can boost

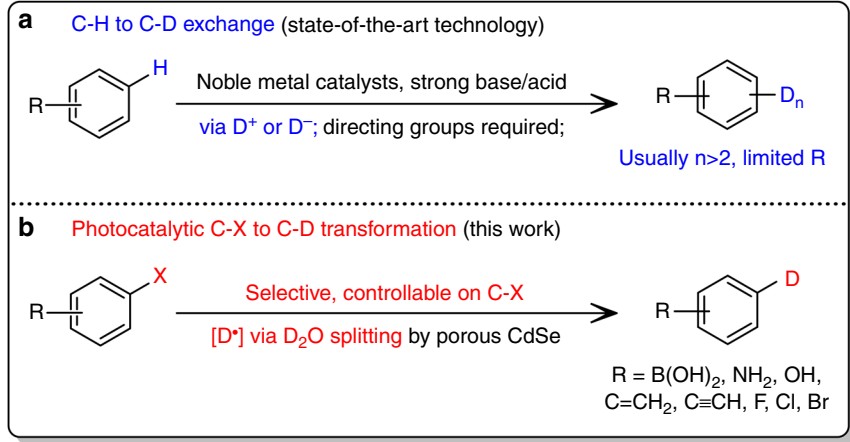

**Fig. 1** Proposed deuteration strategy. **a** State-of-the-art C–H to C–D exchange; **b** photocatalytic C–X to C–D transformation in this work

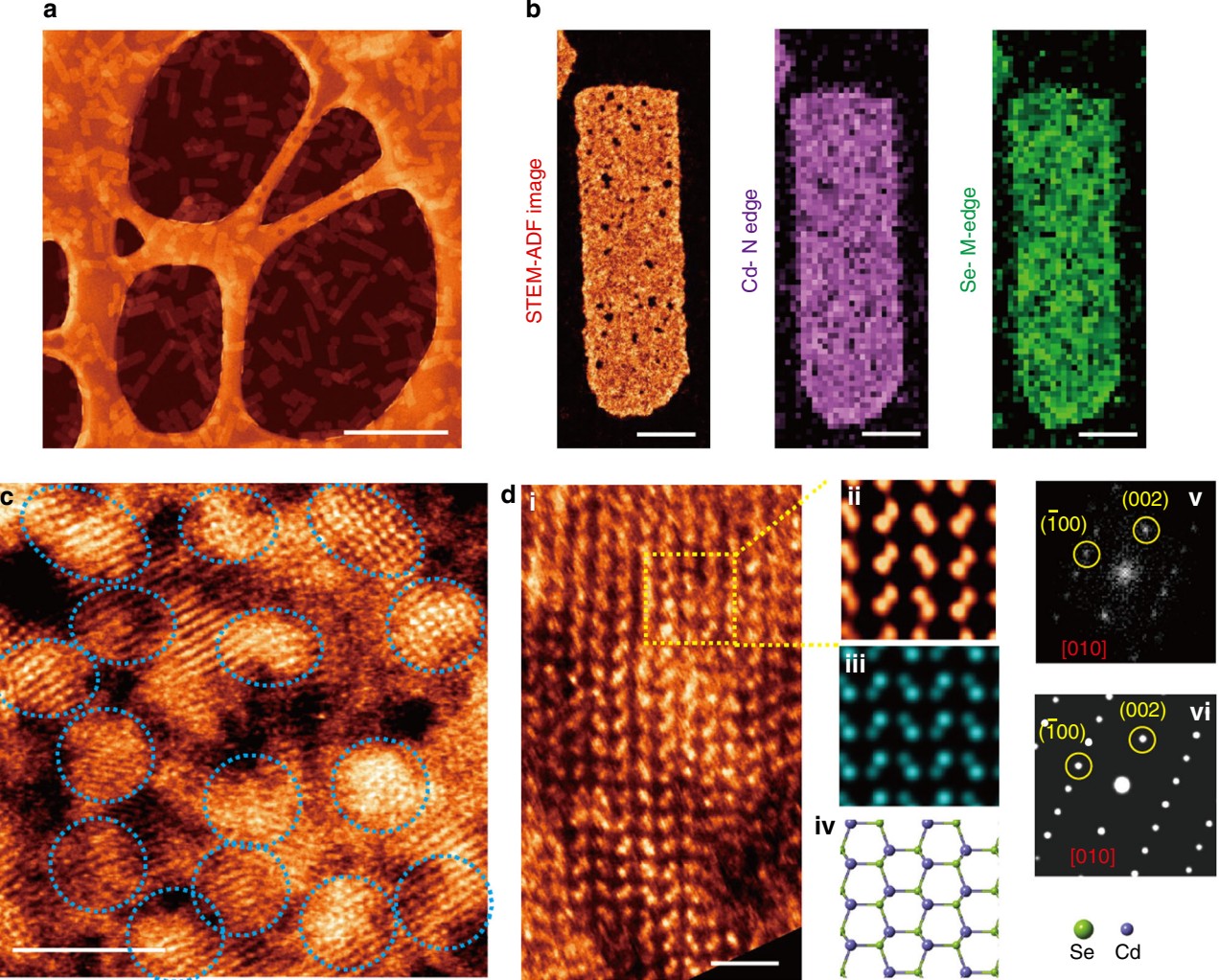

**Fig. 2** Porous CdSe nanosheets. **a** STEM-ADF image of CdSe nanosheets, scale bar: 500 nm; **b** STEM-ADF image and corresponding EELS mapping of CdSe nanosheets, scale bar: 20 nm; **c** atomic resolution STEM-ADF image of CdSe nanocrystalline domains, scale bar: 2 nm; and **d** zooming in on a nanocrystal domain: (ii) enlarged filtered image of the region enclosed by the yellow box in (i) with its corresponding (iii) simulated image, (iv) atomic structure, (v) fast Fourier transform (FFT) pattern, and (vi) simulated FFT pattern, scale bar: 1 nm

catalytic activity owing to their lower coordination numbers and modified electronic structures[31,32].

Ultrathin CdSe nanosheets can be synthesized in bulk quantities by a colloidal method based on previously reported procedure[33]. To improve the catalytic activity, 2D-CdSe nanosheets were corroded under acidic condition to increase porosity. The porous CdSe nanosheets were characterized by scanning transmission electron microscopy in the annular dark field mode (STEM-ADF). As shown in Fig. 2 and Supplementary Fig. 5b, as-synthesized CdSe nanosheets consist of {002} planes; the typical thickness of these sheets as measured by atomic force microscopy (AFM, Supplementary Fig. 4) is 1.7 nm (~seven layers). The porous CdSe nanosheets contain nanocrystalline domains as well as irregular pores, as shown in Fig. 2c, Supplementary Fig. 1–3 and Supplementary Table 3. STEM-ADF image and selected area electron diffraction pattern in Fig. 2d reveal that the CdSe nanosheets have hexagonal wurtzite structure. Electron energy loss spectroscopy elemental mapping and XPS data of Cd and Se (Fig. 2b, Supplementary Fig. 5 & Notes 1 ~ 3) validate that the CdSe nanosheets consists of nanocrystalline phase with a Cd/Se ratio of ~1:0.9.

The porous CdSe nanosheet consists of nanocrystalline domains, which are likely to have its optical properties modified by zero-dimensional quantum confinement[34]; the presence of Cd sites with unsaturated coordination further increases the electron density at the Fermi level and enhances its catalytic properties (Supplementary Fig. 10). Figure 3a compares the UV-Vis spectra of porous and non-porous CdSe nanosheets. The absorption peaks of the lowest energy electron-light hole and electron-heavy hole excitation in porous CdSe are blue-shifted by 12 nm, which are manifestations of the domain size reduction after the chemical treatment to generate porosity[35]. Porous CdSe also shows two red-shifted components in the photoluminescence (PL) spectra, which are absent in the non-porous sample. The red-shifted peaks are assigned to emission from trap or edge sites populated by the non-radiative relaxation of photo-excited electrons, which increases the probability of subsequent electron transfer to chemical species in the solution, thus resulting in approximately three to four times higher $H_2$ evolution rate for our porous CdSe nanosheets (Fig. 3c) compared with commercial CdSe nanoparticles and nanoplates. The highly efficient PHE is the key to achieve successful C–X mediated deuteration, as discussed in the following section[36]. The band edge alignments in porous, 2D-CdSe and other photocatalysts are provided in Fig. 3b and Supplementary Fig. 6.

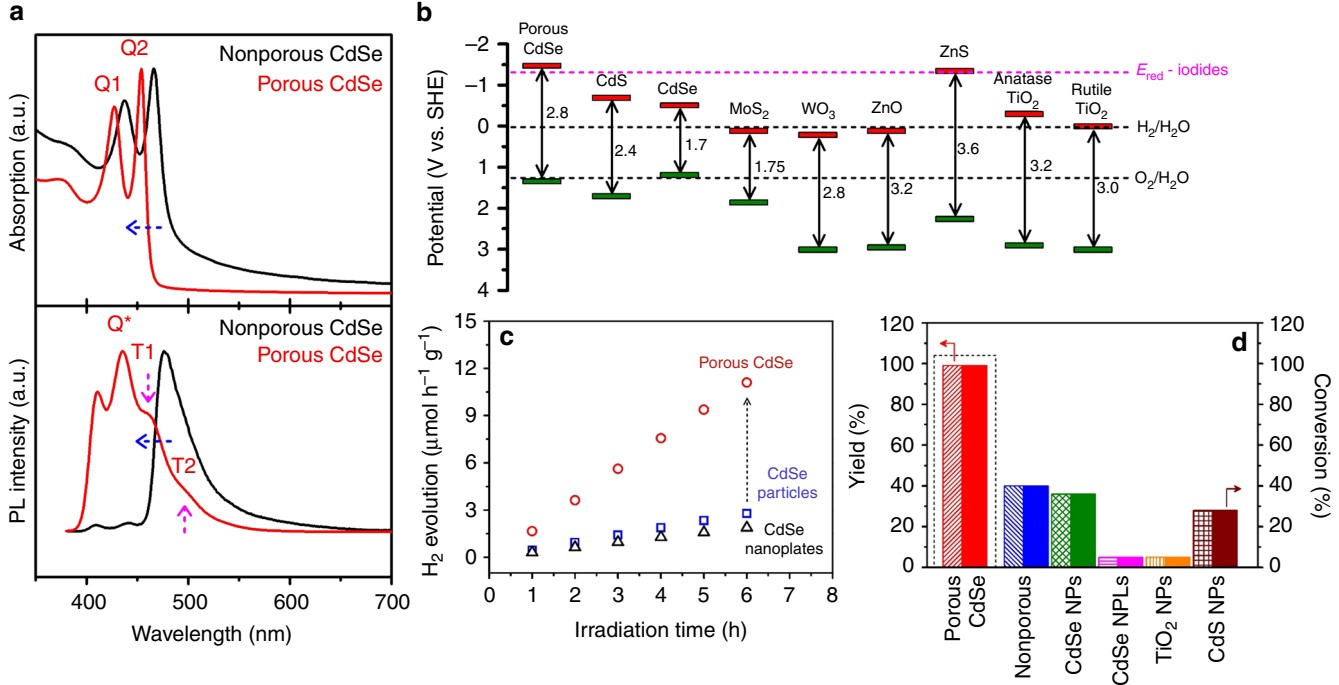

**Fig. 3** Band structure and photocatalytic performance of porous CdSe nanosheets. **a** UV-Vis and PL spectra of porous and non-porous CdSe nanosheets; **b** band edge energies of various photocatalysts compared with redox energy levels for water splitting and for the reduction of aryl iodides; **c** a comparison of the PHE activities of porous CdSe, commercial CdSe nanoparticles, and nanoplates; and **d** comparison of the yields and conversions for the hydrodehalogenation of *p*-iodoanisole using various photocatalysts

**Photocatalytic C–X to C–D transformations**. A core feature of our strategy is using the *in situ* photo-generated D radicals to realize deuterium incorporation in aryl ring through C–X bonds cleavage. The hydrodehalogenation of halides is chosen as the model reaction for studying C–X to C–D transformation. To dissolve the organic substrates, acetonitrile ($CH_3CN$) is used to form a 1:1 mixture with water, in which case nearly quantitative yield is achieved (Supplementary Table 5, entry 1). Isotope labeling by deuterium was performed to determine the origin of H. In the case of $D_2O/CH_3CN$, the reduction product 2ao was obtained in 91% isolated yield with a deuteration ratio of 93% as determined from NMR (Supplementary Fig. 9 (b)), whereas no deuterated product was detected when $H_2O/CD_3CN$ mixture was used as the solvent (Supplementary Fig. 9 (c)). These results prove convincingly that water, rather than $CH_3CN$, is the primary H donor for this C–I hydrogenation reaction. The highest yield was obtained by combining 5 mg of CdSe with 0.25 M of $Na_2SO_3$; complete conversion took place within 2 h (Supplementary Table 5, entry 1). Control experiments using catalysts such as CdSe nanoparticles, CdSe nanoplates, commercial CdS, $TiO_2$ nanoparticles or other sacrificing agents give lower catalytic performance for hydrodehalogenation as compared with our porous CdSe nanosheets. The presence of light, photocatalyst and a sacrificial agent are essential for the hydrodehalogenation reaction (Fig. 3d, Supplementary Tables 5 and 6). Although UV light is sufficient to dehalogenate many organic halides, we did not observe any hydrogenated product in the absence of CdSe photocatalyst (Supplementary Table 4). It is important to note that hydrogen gas cannot be directly used for hydrodehalogenation. No formation of 2c' was detected in the dark in the presence of 1 atm. $H_2$ (all other conditions remaining identical) even after a reaction time of 24 h (Supplementary Table 5, entry 11). Therefore, the hydrogenation of aryl iodide should be owing solely to H radicals generated from photocatalytic water splitting. It should be pointed out that all the substrates tested can be photocatalysed using porous CdSe under visible-light excitation, the reaction time is typically 2 ~ 3 times longer than with UV excitation (see Table 1 and Supplementary Table 5, entry 12 for a comparison of performances using UV and visible light), but comparable yields can be obtained.

To validate the universality of this photocatalytic hydrodehalogenation of halides with water as the hydrogen source, a wide range of halogen-containing substrates were examined. For the sake of investigating the substrate scopes, some experiments were conducted in UV light owing to its shorter reaction time. As listed in Table 1, a wide range of iodinated substrates, including aryl, heteroaryl, alkyl, and alkynyl iodides are amenable to our strategy, producing the corresponding hydrogenated products in good to excellent yields with good functional groups tolerance. Obvious chemoselective hydrogenation of C–I bonds can be seen in the presence of other halogen substituents (F, Cl, and Br) on aryl rings (Table 1, 2n and Table 2, 2aj, 2ak). This chemoselectivity may be attributed to the different reduction potentials and bond dissociation energies of C–X bonds (Supplementary Table 2)[37]. For 1,3,5-triiodobenzene substrate, we can obtain benzene, iodobenzene, and 1,3-diiodobenzene as the major hydrogenated product in a stepwise manner through minor modifications of standard conditions (Supplementary Table 7), which attests to the remarkable flexibility of our methodology. The C–Br, C–Cl, and C–F bonds can be also be hydrogenated in Table 1 and Supplementary Table 7 using longer reaction time.

Based on the optimum conditions of hydrodehalogenation, the substrate scopes for deuteration were investigated with $D_2O$ as deuterium source. As expected, deuteration only occurs at the C–X position with good efficiency, as shown in Table 1. Note that comparable yields are obtained for deuteration and hydrogenation of the same substrate although a longer reaction time is needed for deuteration due to kinetic isotope effect (Table 1). Both the electron-rich and -deficient iodinated substrates can be deuterated by $D_2O$ (Table 1, 2a–2n). The deuterium can be

**Table 1 Scope of photocatalytic C–X to C–D transformation[a]**

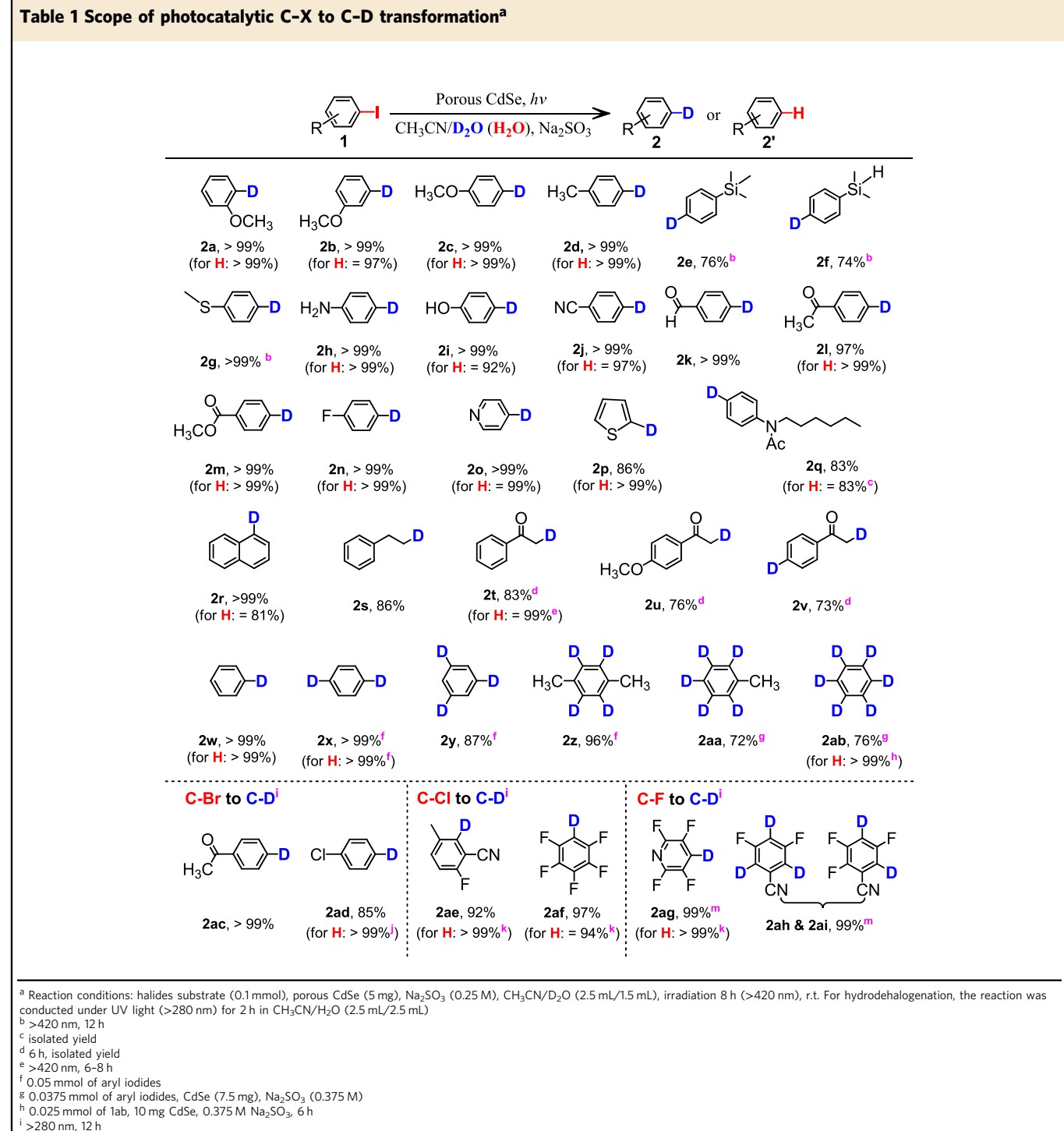

[a] Reaction conditions: halides substrate (0.1 mmol), porous CdSe (5 mg), Na$_2$SO$_3$ (0.25 M), CH$_3$CN/D$_2$O (2.5 mL/1.5 mL), irradiation 8 h (>420 nm), r.t. For hydrodehalogenation, the reaction was conducted under UV light (>280 nm) for 2 h in CH$_3$CN/H$_2$O (2.5 mL/2.5 mL)
[b] >420 nm, 12 h
[c] isolated yield
[d] 6 h, isolated yield
[e] >420 nm, 6–8 h
[f] 0.05 mmol of aryl iodides
[g] 0.0375 mmol of aryl iodides, CdSe (7.5 mg), Na$_2$SO$_3$ (0.375 M)
[h] 0.025 mmol of 1ab, 10 mg CdSe, 0.375 M Na$_2$SO$_3$, 6 h
[i] >280 nm, 12 h
[j] >420 nm, 12 h
[k] >280 nm, 10 h
[m] conversion yields were reported. Yields (Error ca. 5%) were calculated from GC measurements using standard curve

specifically installed at different positions on the aryl ring without negative impact on efficiency (Table 1, 2a–2c). Owing to the mild reaction conditions, sensitive groups such as cyan, ester, amino, hydroxyl, aldehyde, and ketone are not affected (Table 1, 2h–2m), in sharp contrast to current C–H/C–D exchange and early-stage C–X deuteration processes involving strong bases. The latter requires expensive deuterium sources (CD$_3$OD[38], DCOONa[39]), harsh reaction conditions (e.g., BuLi at −78 °C)[40] and is disadvantaged by very narrow substrate scopes. Although the H/D exchange of -NH$_2$ and -OH groups happens in D$_2$O inevitably, the influence on the deuteration of C–I bond is negligible (Table 1, 2h and 2i). Heterocyclic aryl iodides, as well as iodinated substrates containing a long aliphatic chain and naphthyl are good candidates, giving rise to the deuterated products in high yields (Table 1, 2o–2r). The unactivated or activated alkyl iodides are also amenable to our strategy, albeit the former gives a slightly lower yield (Table 1, 2s–2v). In addition, the number of deuterium on aryl ring can be controlled by

**Table 2 D-labeled tools box from photocatalytic C–I to C–D tansformation[a]**

2aj, > 99%
(for **H**: > 99%[b])

2ak, > 99%[c,d]
(for **H**: = 90%[b])

2al, > 99%[c,e]

2am, > 99%

2an, > 99%

2ao, 91%[f]

2ap, 83%[f]

2aq, 67%[g]

2ar, 56%[f]
(for **H**: = 78%[f])

**Suzuki coupling**

2ak + $B(OH)_2$ → Pd(dppf)Cl$_2$, K$_3$PO$_4$, THF/D$_2$O, 80 °C, 20 h → **3**, Isolated yield = 63%   eq. 1

2al + Br → Pd(dppf)Cl$_2$, K$_3$PO$_4$, DMF/D$_2$O, 80 °C, 20 h → **4**, Isolated yield = 67%   eq. 2

**Click reaction**

2am + N$_3$ → Cu(OAc)$_2$·H$_2$O, CH$_3$CN, RT, 12 h → **5**, Isolated yield = 85%   eq. 3

**C-H bond insertion reaction**

2ao + ethyl diazoacetate → CuI, CH$_3$CN, RT → **6**, Isolated yield = 80%   eq. 4

**Tandem deuteration synthesis**

→ CdSe NSs, Na$_2$SO$_3$, CH$_3$CN/**D$_2$O**, > 420 nm → **7**, GC yield = 51%   eq. 5

[a] Reaction conditions: aryl iodides (0.1 mmol), porous CdSe (5 mg), Na$_2$SO$_3$ (0.25 M), CH$_3$CN/D$_2$O (2.5 mL/1.5 mL), irradiation 8 h (>420 nm), r.t. For hydrodehalogenation, the reaction was conducted under UV light (>280 nm) for 2 h in CH$_3$CN/H$_2$O (2.5 mL/2.5 mL)
[b] >280 nm, 2 h
[c] conversion yield was reported
[d] THF/D$_2$O (2.5 mL/1.5 mL)
[e] DMF/D$_2$O (2.5 mL/1.5 mL)
[f] isolated yield
[g] aryl bromide, >280 nm, 12 h. Yields (Error ca. 5%) were calculated from GC-MS using standard curves

halogen chemistry to give mono-, bi-, or multi-deuterated products, which is extremely challenging in C–H/C–D exchange (Table 1, 2w–2ab). In addition, C–Br, C–Cl, and C–F bonds which are usually non-labile using other photocatalytic methods can be deuterated in the presence of some activated functional groups (Table 1, 2ac–2ai).

In addition to the direct deuterium incorporation on target molecules, the heavy water D$_2$O-splitting strategy proposed here can be developed into a series of useful D-containing tool kits bearing both deuterium atom and linkage moieties. For example, we have successfully incorporated deuterium on aryl halogen, boric acid, alkyne, and alkene moieties-containing molecules (Table 2, 2aj–2ar), which are often fragile under harsh conditions as well as noble metal catalytic system. The elegance of the

approach is that deuterium can be readily incorporated on pharmaceutical molecules or advanced materials through subsequent Suzuki coupling, Click reaction, Heck reaction, or C–H bond insertion reaction (Table 2, eq. 1 ~ 4), which is challenging by current methods. Potentially useful deuterated molecules can be synthesized by our strategy. For example, using the kinetic isotope effect, a deuterium atom at the ortho-position of an amide group is helpful for probing reaction mechanisms in metal-catalyzed transformations, or for determining the reaction sites of a reaction (Table 2, 2ap). By converting C–Br bond to C–D bond, the deuterium labeled pharmaceutical precursor 2aq is synthesized, which can undergo further ester hydrolysis to form D-containing nicotinic acid. Nicotinic acid is a benchmarked anti-hypercholesterolemia drug and among the top 200

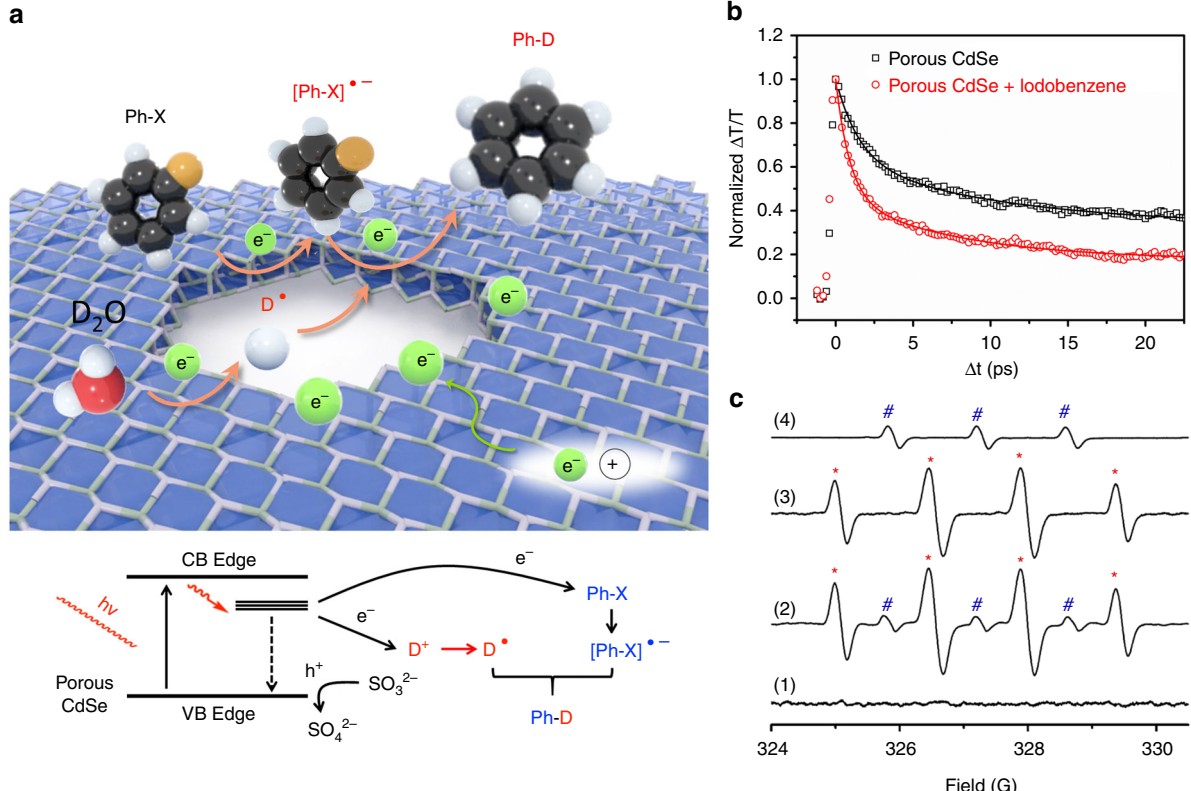

**Fig. 4** Mechanism of the photocatalytic deuteration reaction. **a** possible radical pathway for C–X to C–D transformation; **b** comparison of femtosecond transient absorption kinetics of porous CdSe with/without the addition of iodobenzene in CHCl$_3$. Pump: 400 nm, probe: 450 nm; **c** EPR measurements of 1 mM CdSe nanosheets with 4 mM DMPO and 0.13 M Na$_2$SO$_3$ in (1:1 v/v) CH$_3$CN/H$_2$O under UV (>280 nm) irradiation: (1) with 2 mM $p$-iodoanisole in the dark, (2) with 2 mM $p$-iodoanisole after 30 s irradiation, (3) without $p$-iodoanisole after 35 min irradiation, and (4) only with 1 mM CdSe and 4 mM DMPO in iodobenzene after 150 s irradiation; * and # represent peaks from DMPO-OH and DMPO-C radicals

pharmaceutical products produced in 2009[41]. The halogen-containing amino acids can also be deuterated via the C–X transformation (2ar). Through polymerase chain reaction, deuterium can be integrated into polypeptides, protein etc. for investigating their property owing to the isotope effect. The utility of this deuterated strategy is further demonstrated by a tandem deiodination-cyclization-deuteration of 2-(allyloxy)-1-iodobenzene, from which a radical pathway for the deuteration of halides may be operational (Table 2, eq. 5)[26]. The success in the preparation of deuterated benzofuran scaffold opens up the window on the follow-up synthesis of more complex deuterium-bearing molecules with new stereocentres through radical addition or coupling. Therefore, C–X to C–D transformation can provide a powerful tool kit for deuterium labeling in synthesis and drug investigation.

To illustrate the robustness and scalability of our approach, a gram-scale hydrodehalogenation of 1w (1.53 g) was carried out with a yield of 71% after irradiation for 16 h at r.t. (Supplementary Fig. 7). Moreover, porous CdSe nanosheets can be recycled and maintain a reaction efficiency of >90% after 3 runs. A slight reduction of the reactivity was observed in the fourth run (Supplementary Fig. 8), which may be due to mild photocorrosion and/or loss of catalyst during recovery and washing process[42]. The excellent arylamine group tolerance (Tables 1, 2h) can be attributed to the competition with the sacrificial agent (Na$_2$SO$_3$). As Na$_2$SO$_3$ has an apparently lower oxidation potential than the valence band potential of CdSe and the oxidation potential of arylamine[43], the photo-generated holes will be preferentially transferred to the sacrificial agent, which is in

excess, and not to the arylamine. For alkenes and alkynes (Table 2, 2am & 2an), the super high selectivity is due to the lack of noble metal catalysts for activation[44], as well as the kinetically unfavorable radical addition compared to the fast radical coupling.

**Reaction mechanism for photocatalytic C–X to C–D transformations.** The photocatalytic C–X bond deuteration by D$_2$O involves a radical pathway and a possible mechanism is proposed in Fig. 4a. The reaction begins with the absorption of substrates on the photocatalysts[45]. Upon irradiation, electron-hole pairs are generated on porous CdSe[29,30,46]. Owing to the defective sites, besides electron-hole recombination, electrons are trapped and transferred to D$_2$O and/or organic halides to form the corresponding deuterium radicals[47] or the aryl radical anion (ArX·$^-$)[26–28]. Following, the coupling of deuterium radical and ArX·$^-$ leads to the formation of deuterated product. The D radicals can also self-couple to release D$_2$[42,47]. The catalyst then recovers to its ground state after accepting an electron from the sacrificial agent (Na$_2$SO$_3$), whereupon SO$_3^{2-}$ converts to SO$_4^{2-}$ owing to oxidation[48]. The electron transfer from the CdSe to iodobenzene is evidenced by PL and femtosecond transient absorption (TA) spectroscopy. We observe a significant quenching of PL from CdSe, together with a faster decay in the bleaching band at 455 nm in TA spectra upon the addition of iodobenzene (Fig. 4b and Supplementary Fig. 15–18). A shorter electron transfer time constant was obtained from single-wavelength dynamics at 450 nm in the presence of iodobenzene, as compared to when it was absent.

The radical pathway for photocatalytic deuteration is confirmed by electron paramagnetic resonance (EPR) measurements in Fig. 4c and Supplementary Fig. 11–14. By trapping radicals with 5,5-dimethyl-1-pyrroline-N-oxide (DMPO) molecules, which forms stable radical adducts, the characteristic EPR signals of DMPO-OH adducts (quartet peaks, $a_N = a_H = 14.9$ G, marked by *) can be detected as water/hydroxides are oxidized to hydroxyl radicals by photo-generated holes[49]. Although we could not detect the presence of hydrogen radicals due to trace amounts of oxygen in the system, hydroperoxyl adducts (DMPO-OOH, marked by △) could be detected[50], which might be generated from the combination of hydrogen radicals with trace amount of oxygen in the system. Three new peaks that can be assigned to the reduction product of DMPO-C adducts ($a_N = a_H = 13.9$ G, marked by #) are clearly seen after the addition of iodides[51]. The presence of free radicals in this system is further confirmed by the exponential decay of the EPR peak area with time (Supplementary Fig. 11) after a radical scavenger, 2,2,6,6-tetramethyl-1-piperidinyloxy (TEMPO) is added. Supplementary Fig. 14 compares the EPR results for porous and non-porous CdSe nanosheets. The absence of EPR signals from DMPO-OH and DMPO-C adducts in non-porous CdSe nanosheets points to the much lower photocatalytic activity of this material compared to its porous counterpart.

In conclusion, using porous 2D-CdSe nanosheets as photocatalysts, selective and efficient C–X bond deuteration can be achieved with $D_2O$. A wide range of organic iodides including aryl-, alkyl-, and alkynyl iodides can be effectively deuterated in excellent yields and with good functional groups tolerance. This method is also powerful enough to deuterate less reductively labile C–Br, C–Cl and even C–F bonds in the presence of activating groups on the ring. The good performance is due to high reductive potential of porous CdSe nanosheets and its high density of catalytic sites. This strategy introduces spatially precise and controllable number of deuterium atoms on the substrates compared to current C–H/C–D exchange. Importantly, we can further develop a D-labeled tool kit including deuterated aryl halogen, boric acid, alkyne, alkene, etc., which are useful D-containing building blocks for drugs or advanced materials. The radical coupling strategy can be further exploited to rapidly generate molecular complexity via radical cascade cyclization, for example, for ring closure and the generation of new stereo centers, or in the total synthesis of complex molecules.

## Methods

**Synthesis of porous CdSe nanosheets**. We used a modified method from Ref. [33] to introduce nanopores on CdSe nanosheets. In brief, 1.5 mmol of $CdCl_2$ was dissolved in 5 mL of oleylamine and 5 mL of octylamine at 120 °C for 2 h under Ar. The solution was mixed with 4.5 mmol of selenide dispersed in 2.5 mL of oleylamine and 2.5 mL of octylamine at r.t. The resulting mixture was heated to 100 °C for 16 h under Ar. After precipitation in ethanol and re-dispersing in $CHCl_3$, as-formed CdSe nanosheets were purified using a silica column. The acidic environment of silica column causes the corrosion of CdSe to form nanopores. The porous CdSe nanosheets were then recovered by rotary evaporation.

**Photocatalytic C–X to C–D transformation**. Typically, 5 mg of porous CdSe nanosheets, 0.1 mmol of 4-iodoanisole (1c) and $Na_2SO_3$ (0.25 M) were dispersed in a $CH_3CN/D_2O$ (2.5 mL/1.5 mL) mixture and sonicated for 30 min. The reaction mixture was then irradiated with a Xenon lamp visible light (150 W, $\lambda > 420$ nm) for 8 to 10 h at r.t. under Ar. After reaction, the mixture was centrifuged to remove photocatalyst. The supernatant was extracted by adding 5 mL of $CH_2Cl_2$ and the organic phase was analyzed by GC-MS. The conversions and yields were calculated from standard calibration curves. The yield was calculated by dividing the amount of the obtained desired product by the theoretical yield. Hydrogenated products were prepared by using $CH_3CN/H_2O$ mixture under identical conditions. Details on reaction setup and synthesis of substrates can be found in Supplementary Fig. 19 ~ 76 and Supplementary Methods.

**Characterization equipment**. STEM and EELS (Nion UltraSTEM-100 with aberration-correction, 60 kV), TEM/EDS (FEI Titan, 80 kV), AFM (Dimension Fast Scan), XPS (AXIS UltraDLD, monochromatic Al $K_a$), XRD (Bruker D8), NMR (Bruker AV300), EPR (JEOL FA200), GC (Agilent 7890 A), GC-MS (Agilent 5975 C inert MSD with triple-axis detector), UV-Vis (Shimadzu UV-3600), FT-IR (Varian 3100), PL (Horiba Fluorolog-3), TA & Pump-Probe (Spectra Physics, Ti: sapphire femtosecond laser), Electrochemistry (Autolab PGSTAT30 with a 3-electrodes cell).

**Data availability**. All data are available from the authors upon reasonable request.

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

## Acknowledgements

K.P. Loh thanks the National Research Foundation, Prime's Minister Office, for support under the mid-sized research centre (CA2DM), and also MOE Tier 2 grant "Porous, Conjugated Molecular Framework for Energy Storage (MOE2016-T2-1-003)." C. Su thanks NNSFC (51502174), Shenzhen Peacock Plan (Grant No. 827-000113, KQTD2016053112042971), and Science and Technology Planning Project of Guangdong Province (2016B050501005) for financial support. We acknowledge Dr. Ji'en Wu for his contributions on EPR measurements, Dr. Tobias Lünskens, Reinhard Haselberges, Dr. Dong Shi, Dr. Bo Liu, and Mr. Xing Li for PL measurements, Ms. Qianqian Hu for XPS measurements, Prof. Bin Zhang and Dr. Wei Meng for insightful discussions. This work was supported in part by the U.S. Department of Energy, Office of Science, Basic Energy Science, Materials Sciences and Engineering Division, and through a user project at ORNL's Center for Nanophase Materials Sciences (CNMS), which is a DOE Office of Science User Facility. C.L. appreciates funding support from Shenzhen University and the China Postdoctoral Science Foundation (2016M592519). Z.C. thanks the NGS Scholarship for support.

## Author contributions

C.L., Z.C. conceived the research and wrote the draft. C.L. and Z.C. synthesized the materials and C.L. performed the photocatalytic reactions. X.Z., Z.C., and W.Z. conducted TEM characterization and data analysis. G.N., C.L., and Z.C. conducted EPR measurements. H.Z., C.L., Z.C., and Q.X. performed the transient measurements. K.L. performed AFM measurement. Q.G., W.T., W.F., B.T., X.P., and J. L. assisted with materials characterization and data analysis. C.S. and K.P.L supervised the research. All authors discussed and commented on the manuscript.

## Additional information

**Competing interests:** The authors declare no competing financial interests.

