## [Peer Review File · Nature Communications]

Reviewers' comments:

Reviewer #1 (Remarks to the Author):

The authors report the C-I to C-D bond transformation catalyzed by heterogeneous visible light photocatalysis. The project is well described and particularly the material characterization is detailed.

However, I do not agree with the motivation of the work, which aims at simplifying deuteration of organic molecules. The problem of selective functionalization is moved to the iodination step. The catalytic dehalogenation can be achieved with many transition metal catalyzed or photocatalytic processes, some of which were cited by the authors. The reported procedure does not provide an advantage compared with known methods. The value of the report is in the material characterization, but this does not justify an urgent publication for a broader audience. The mechanistic proposal requires revision. A reaction of a radical anion with a deuterium radical is proposed. The radical anion of aryl iodides is not stable and expels the iodide anion immediately. It is very unlikely that two highly reactive radical species, an aryl radical and the deuterium radical, combine. The term photocatalytic water splitting is also misleading, as sulfite is used as a sacrificial electron donor.

Overall, this is a rather special application of CdSe photochemistry. The impact for synthesis is small and the mechanistic proposal requires revision. The work may be publishable in a specialized journal focusing on material aspects after thorough revision.

Reviewer #2 (Remarks to the Author):

The manuscript by Loh *et al.* reports on a photo-induced reductive hydrodehalogenation of organic halides using porous 2D-CdSe nanosheet as photocatalyst from water or D₂O as H or D sources. Deuterated compounds are highly important for the metabolic analysis of bioactive agents as well as the mechanistic study of chemical and enzymatic reactions, therefore this discovery of a practical deuteration method is very important.

Overall, the reaction appears general as it proceeds well for a number of different substrates. First, C-X to C-H transformation was explored regarding different aryl-, alkyl-, alkynyl iodides as coupling partners (22 examples, Figure S6). The scope was then successfully extended to other R-X (X = Br, Cl, F) (12 examples, Figure S7). In a second part, C-X to C-D transformation

was explored by replacing H₂O to D₂O. Similar scope was investigated with the deuteration of aryl halides, as well as alkyl halides (39 examples, Table 1 and 2). Interestingly, the reaction is very chemoselective and a wide range of functional groups is tolerated, including some sensitive one such as C–X and C–B(OH)₂ bonds, which could be easily converted in more complex molecules. A reasonable radical pathway is proposed based on mechanism investigation including electron paramagnetic resonance (EPR) measurements and trapping radical experiments.

1. This reference about palladium-catalytic deuterodechlorination of aryl/heteroaryl chlorides should be added (J. Org. Chem. 2016, 81, 8934–8946).
2. Abbreviation for acetonitrile should be CH₃CN and not AN.
3. The first part about C–X to C–H transformation is difficult to follow as all scheme and table are in SI. I strongly suggest to include these data in the main document. These can be combine with the Table 1 and Table 2 with two entries H and D.
4. What is the stability of silicon reagent under these reaction conditions? Deuteration of 1-bromo-4-(trimethylsilyl)benzene or (4-bromophenyl)methylphenylsilane could be investigated.
5. What is the reactivity of bromothioanisole or iodothioanisole under this reaction conditions.
6. What is the reactivity ortho-dibromobenzene?
7. Alkenyl halides could be deuterated?
8. The authors should explain why the reaction is chemoselective regarding different C–X bonds. Indeed, the optimized conditions allow the deuteration of aryl bromides, and aryl chloride but when 1-bromo-4-iodobenzene was used as substrate only C–I bond was touched.

These results, which are accurately documented within the Supporting Information, are impressive and set a new benchmark for the specific deuteration, particularly within challenging aliphatic chains. Therefore, I recommend its publication in nature communication after answer to the above comments.

Reviewer #3 (Remarks to the Author):

This communication reports an elegant approach for deuterium labelling through photoredox reactions catalysed by newly designed porous semiconducting (CdSe) nanosheet and using heavy water as deuteration agent to convert C–X to C–D efficiently and selectively, which shows distinct mechanism compared to traditional C–H/C–D exchange strategy. The reviewer believes that this method would significantly inspire new ideas of production of deuterated compounds in organic chemistry, which will thus impact both academic research and industrial manufacturing.

The data and methodology is adequate to support the conclusion. The figures and tables are organized well.

However, the following minor comments need to be addressed to further improve the quality of this communication:

- 1) The effect of the sacrificing agent (additive) as electron donor on the catalytic performance should be discussed more in the text. Only two additives, Na₂SO₃ and OA were used in this study. What about other popular additives like triethylamine, etc?
- 2) It is surprising to see the super high selectivity in this transformation, such as arylamine, alkene and alkyne, which are generally reactive in photoredox catalysis. It is necessary to elaborate more from a mechanistic perspective, and give the specific reaction conditions if necessary.
- 3) Supplementary Table 5 is a little bit heavy. Actually some entries are not essential to display. For examples, Entry 1 using water only did not give any product because of solubility issue of the halo-compound, which has been mentioned in the text. Entries 3-5 are longer time reaction results for Entry 6. There is no difficulty to understand but easy to confuse the readers.
- 4) The reaction setup is missing. Considering the experimental reproduction from others, it is important to provide the specification, as well as light intensity or supply power for UV light source.

Additionally, one recent review (chem Soc Rev 2016 (45), 6165-6212) comprehensively described the photoredox catalysis for organic and polymer synthesis. It is suggested to consider to cite this article.

Point-by-point responses to reviewers' questions for "Controllable Deuteration of Halogenated Compounds by Photocatalytic D₂O Splitting" (NCOMMS-17-22821).

Reviewer #1:

The authors report the C-I to C-D bond transformation catalyzed by heterogeneous visible light photocatalysis. The project is well described and particular the material characterization is detailed.

However, I do not agree with the motivation of the work, which aims simplifying deuteration of organic molecules. The problem of selective functionalization is moved to the iodination step. The catalytic dehalogenation can be achieved with many transition metal catalyzed or photocatalytic processes, some of which were cited by the authors. The reported procedure does not provide an advantage compared with known methods. The value of the report is in the material characterization, but this does not justify an urgent publication for a broader audience.

Response: We appreciate the comments. We have to reiterate that our C-X to C-D transformation offers significant advantages compared to current C-H to C-D exchange and traditional C-X to C-D transformation. The reasons are explained as follow (detail comparisons in Supplementary Table 1):

1) The comment, "*The problem of selective functionalization is moved to the iodination step*" has to be viewed in context. Taking advantage of well-established halogenation chemistry, we can achieve **selective deuteration, which is one of the major challenges** in current C-H to C-D exchange, where noble metal catalysts, harsh reaction conditions *etc* are typically used to activate the C-H bonds for subsequent deuteration (*Angew. Chem. Int. Ed.* **2007**, 46, 7744-7765).

Although our approach requires selective iodination (halogenation) as the prerequisite, **the activation of C-I (C-X) bonds is principally much easier than C-H bonds** due to lower bonding energy. This is important because the C-X to C-D transformation can be achieved in **a controllable manner under mild reaction condition** with **much higher yield and selectivity** than current C-H to C-D exchange. The well-developed halogen chemistry also ensures the efficient pre-installation and de-installation of halogen on desired molecules. Actually, C-I (C-X) naturally exists in many biomolecules and industrial products, thus pre-iodination (halogenation) is NOT NECESSARY in some cases (*Chem. Rev.* **2017**, 117, 5619-5674). A good example will be the deuteration of halogen-containing amino acid (**2ar** in Table 2).

2) The comment, "*The catalytic dehalogenation The reported procedure does not provide an advantage compared with known methods*" is **not correct**. The traditional procedures for hydrodehalogenation CANNOT be simply utilized for deuteration because **the deuteration of hydrogen donors** (amines, alcohols, or Hantzsch esters *etc*) is very challenging.

To the best of our knowledge, most catalytic deuterio-dehalogenation are conducted in harsh reaction conditions (*e.g.*, Grignard reagents or n-BuLi at -78 °C, *Chem. Rev.* **2002**, 102, 4009-4091) or employ noble metals catalysts (*Org. Chem. Front.* **2015**, 2, 1071-1075), with a very limiting substrate scope. Despite some recent advances in this area, such as the deuterodechlorination using Pd-NHC complexes mentioned by Reviewer 2 (*J. Org. Chem.* **2016**, 81, 8934-8946), the yield and selectivity of C-X to C-D transformation is still far from satisfactory.

We solve this dilemma by using **heavy water (D₂O) as the deuteration reagent**, which offers distinct advantages in terms of safety, cost and handling. Since the O-D bonds have quite high dissociation energy, we carried out photocatalytic hydrogen evolution (PHE) by a non-noble catalyst (porous CdSe) to generate

active D radicals from D₂O for organic deuteration. The benefit of our approach is that it is mild, green and effective, which can achieve **excellent selectivity and functional group tolerance** (See our response to Reviewer 3, Question 2).

Therefore, we are able to develop a **deuterium-labelled tool kit**, including deuterated boronic acids, halides, alkynes, alkenes, aldehydes *etc* due to the excellent functionalities tolerance of our strategy (Main Text Table 2). Note that most of these deuterium-labelled molecules are incompatible with current methods, we can further explore this tool kit for the synthesis of deuterated pharmaceutical molecules or advanced materials through Suzuki coupling, Click reaction, Heck reaction, C-H bond insertion reaction *etc*; This is the **first demonstration** of photo-generated deuterium radicals from heavy water to produce valuable chemicals.

In addition, our C-X to C-D approach involves a **deuterium radical (D•) pathway**, which is mechanistically distinct from the current methods involving deuterium cation (D⁺) or anion (D⁻), this allows us to rapidly generate molecular complexity by radical coupling or cyclization. Here we provide an example of tandem intramolecule deiodination-cyclization-deuteration to synthesize deuterated benzofuran derivatives (Table 2, eq. 5), which is impossible by current C-H/C-D exchange and traditional C-X to C-D transformation.

3) The comment, “*The value of the report is in the material characterization, but this does not justify an urgent publication for a broader audience*” underestimates the significance of our materials engineering on porous CdSe.

Rational design of photocatalytic system, especially the fine tuning of photocatalyst performance by modifying its structure or morphology, has been an important area of research for the last decade. Many engineered photocatalyst tested on standard reactions have been published in *Nature* related journals. The main challenge for C-X to C-D transformations is the lack of a suitable photocatalyst with a **reduction potential more negative than organic halides**. Common photocatalysts such as TiO₂, ZnO, bulk CdSe *etc* failed to achieve this goal, as shown in Main Text Figure 3b. It is even more challenging to achieve this when visible light is used as the photoexcitation source (ZnS is ruled out as it has a bandgap of ~ 3.6 eV).

We are able to achieve such high photocatalytic efficiency by engineering the morphology of CdSe. First we synthesized 2D CdSe nanosheets, next we introduced a high density of nanopores in our 2D CdSe nanosheets. We found that the introduction of nanoporosity on 2D CdSe downshifts the conduction band to ~ - 1.6 V vs. SHE to match the reduction potential of halides, while the band gap remains in the appropriate region (~ 2.8 eV) for visible light catalysis. A high density of active Cd sites on porous CdSe further boosts the catalytic performance for deuteration. The engineering of catalyst to target high performance catalysis represents significance advance and deserves to be published.

Overall, we believe our findings represent a major advance in the area of deuterium chemistry and should be disseminated rapidly in *Nature Communications* to the large research community working on photo-organic chemistry and 2D materials.

The mechanistic proposal requires revision. A reaction of a radical anion with a deuterium radical is proposed. The radical anion of aryl iodides is not stable and expels the iodide anion immediately. It is very unlikely that two highly reactive radical species, an aryl radical and the deuterium radical combine. The term photocatalytic water splitting is also misleading, as sulfite is used as sacrificial electron donor.

Response: Thank you for the criticisms on the reaction mechanism. The EPR results in Main Text Figure 4c and Supplementary Figures 11 - 14 confirm the existence of photo-generated H and C radicals in our system. Although the radical anion of aryl iodides is not stable, they can react with deuterium radicals prior to the decomposition into non-reactive species, the reasons are provided as follow:

1) Our C-X to C-D transformation is mediated by porous CdSe catalyst (i.e., surface reaction), which means that the local concentration of both radicals are far higher than in bulk solution and provides a high possibility for them to recombine near the CdSe surface. A good example is given by Kisch *et al.* (*Chem. Eur. J.* **1999**, 5, 208-217; *Acc. Chem. Res.* **2017**, 50, 1002-1010), where an unexpected chemo-selectivity is found at the solid-liquid interface in hetero-catalytic reactions compared to those in the bulk solution;

2) In practice, D radical (D₂O) is in excess amount. The reaction kinetics for water splitting is also faster than the cleavage of organic halides. So that the radical anion of aryl iodides, once generated, will quickly react with the nearby D radicals to form the deuterated compounds;

3) The coupling between radical anions and radicals is common in photocatalytic organic transformations (*Science* **2011**, 334, 1114-1117; *J. Am. Chem. Soc.* **2015**, 137, 13768-13771; *Nat. Chem.* **2017**, 9, 453-456 & *Chem. Soc. Rev.* **2017**, 46, 5193-5203.). Although the radical anion of aryl iodides is not stable in nature, it will first be converted into reactive aryl radicals instead of non-reactive species, as proposed by many literature involving photocatalytic iodide transformations (to name a few, *Chem. Rev.* **2013**, 113, 5322-5363; *Acc. Chem. Res.* **2016**, 49, 2295-2306 & *Org. Chem. Front.* **2016**, 3, 1011-1027). The recombination between aryl radicals and D radicals can also yield the corresponding deuterated compounds and does not affect the overall reaction.

Regarding the term, “photocatalytic water splitting”, we believe it is not misleading even though sulfite is used as the sacrificial electron donor. This term is defined as “*the photocatalytic water splitting into H₂ and O₂, and H₂ or O₂ evolution from an aqueous solution containing a sacrificial reagent*” (Heterogeneous photocatalyst materials for water splitting, *Chem. Soc. Rev.* **2009**, 38, 253-278). In this report, porous CdSe serves as a dual-functioning catalyst for both water splitting (or more accurate, H₂/D₂ evolution) and organic transformations. We can also observe and monitor the generation of H₂ gas in Main Text Figure 3c. So we believe this term is appropriate for our study.

Overall, this is a rather special application of CdSe photochemistry. The impact for synthesis is small and the mechanistic proposal requires revision. The work may be publishable in a specialized journal focusing on material aspects after thorough revision.

Response: We believe we have addressed your concerns on the novelty and the reaction mechanism.

Reviewer #2:

*The manuscript by Loh *al.* reports on a photo-induced reductive hydrodehalogenation of organic halides using porous 2D-CdSe nanosheet as photocatalyst from water or D₂O as H or D sources. Deuterated compounds are highly important for the metabolic analysis of bioactive agents as well as the mechanistic study of chemical and enzymatic reactions, therefore this discovery of a practical deuteration method is very important.*

Overall, the reaction appears general as it proceeds well for a number of different substrates. First, C-X to C-H transformation was explored regarding different aryl-, alkyl-, alkynyl iodides as coupling partners (22 examples, Figure S6). The scope was then successfully extended to other R-X (X = Br, Cl, F) (12 examples, Figure S7). In a second part, C-X to C-D transformation was explored by replacing H₂O to D₂O. Similar scope was investigated with the deuteration of aryl halides, as well as alkyl halides (39 examples, Table 1 and 2). Interestingly, the reaction is very chemoselective and a wide range of functional groups is tolerated, including some sensitive one such as C-X and C-B(OH)₂ bonds, which could be easily converted in more complex molecules. A reasonable radical pathway is proposed based on mechanism investigation including electron paramagnetic resonance (EPR) measurements and trapping radical experiments.

Response: We have addressed the referee’s concerns on the reactivity of halogenated silicon reagents, thioanisoles, dibromobenzene and alkenyl halides *etc.* We also combine the Supplementary Tables 6 & 7 into Main Text Tables 1 & 2 according to the suggestions. Related discussions on the chemo-selectivity have been included in the revised version.

Question 1: *This reference about palladium-catalytic deuterochlorination of aryl/heteroaryl chlorides should be added (J. Org. Chem. 2016, 81, 8934–8946).*

Response: We have already cited and commented on this reference in the revised version:

Page 3, Line 18: “For example, Onomura group developed a Pd-NHC system for effective deuterochlorination of aryl/heteroaryl chlorides²¹.”

Question 2: *Abbreviation for acetonitrile should be CH₃CN and not AN.*

Response: We have revised all improper abbreviations (AN) into “CH₃CN” throughout the manuscript.

Question 3: *The first part about C-X to C-H transformation is difficult to follow as all scheme and table are in SI. I strongly suggest to include these data in the main document. These can be combine with the Table 1 and Table 2 with two entries H and D.*

Response: Thank you so much for the suggestion. We agree that the original version of the C-X to C-H transformation is difficult to follow and we have combined the related scheme and tables into the Main Text Tables 1, 2 & Supplementary Table 7, see below:

Table 1 | Scope of photocatalytic C-X to C-D transformation ^a

C-Br to C-D ⁱ

C-Cl to C-D ⁱ

C-F to C-D ⁱ

^a Reaction conditions: halides substrate (0.1 mmol), porous CdSe (5 mg), Na₂SO₃ (0.25 M), CH₃CN/D₂O (2.5 mL/1.5 mL), irradiation 8 h (> 420 nm), r.t.; For hydrodehalogenation, the reaction was conducted under UV light (> 280 nm) for 2 h in CH₃CN/H₂O (2.5 mL/2.5 mL); ^b > 420 nm, 12 h; ^c isolated yield; ^d 6 h, isolated yield; ^e > 420 nm, 6-8 h; ^f 0.05 mmol of aryl iodides; ^g 0.0375 mmol of aryl iodides, CdSe (7.5 mg), Na₂SO₃ (0.375 M); ^h 0.025 mmol of 1ab, 10 mg CdSe, 0.375 M Na₂SO₃, 6 h; ⁱ > 280 nm, 12 h; ^j > 420 nm, 12 h; ^k > 280 nm, 10 h; ^m conversion yields were reported. Yields (Error ca. 5%) were calculated from GC measurements using standard curve.

Table 2 | D-labelled Tools Box from Photocatalytic C-I to C-D Transformation^a

Suzuki coupling

Click reaction

C-H bond insertion reaction

Tandem deuteration synthesis

^a Reaction conditions: aryl iodides (0.1 mmol), porous CdSe (5 mg), Na₂SO₃ (0.25 M), CH₃CN/D₂O (2.5 mL/1.5 mL), irradiation 8 h (> 420 nm), r.t.; For hydrodehalogenation, the reaction was conducted under UV light (> 280 nm) for 2 h in CH₃CN/H₂O (2.5 mL/2.5 mL); ^b > 280 nm, 2 h; ^c conversion yield was reported; ^d THF/D₂O (2.5 mL/1.5 mL); ^e DMF/D₂O (2.5 mL/1.5 mL); ^f isolated yield; ^g aryl bromide, > 280 nm, 12 h. Yields (Error ca. 5%) were calculated from GC-MS using standard curves.

Supplementary Table 7. Other Functional Substrates for Hydrogenation

Iodides ^a

Bromides ^c

Chlorides ^c

Fluorides ^c

Stepwise hydrogenation:

^a Reaction conditions: aryl iodides (0.1 mmol), porous CdSe (5 mg), Na₂SO₃ (0.25M), CH₃CN/H₂O (2.5 mL/2.5 mL), irradiation 2 h (> 280 nm), r.t.; ^b > 420 nm, 6-8 h; ^c > 280 nm, 10 h; ^d Conversion yields were reported; ^e 0.05 mmol of 1,3,5-triiodobenzene, DMF/H₂O (2.5 mL/2.5 mL), 2 h; ^f 4.5 h; ^g 8 h. Yields (Error ca. 5%) were calculated from GC-MS using standard curves.

Question 4: *What is the stability of silicon reagent under these reaction conditions? Deuteration of 1-bromo-4-(trimethylsilyl)benzene or (4-bromophenyl)methylphenylsilane could be investigated.*

Response: According to the reviewer's suggestion, we have examined the deuteration of brominated silicon reagents in Figure 1. Briefly, 1-bromo-4-(trimethylsilyl)benzene is stable in reaction condition, which gives 48 % yield of deuterated product after UV irradiation (> 280 nm) for 8 h (other conditions are identical to the *Supplementary Methods*). Although we are not able to obtain (4-bromophenyl)methylphenylsilane at current stage, we synthesize another similar silicon reagent ((4-bromophenyl)dimethylsilane) and examine its reactivity (GC yield = 40 %, UV, 8 h).

To further evaluate the applicability of silicon reagent, we also synthesize and examine the corresponding iodinated silicon reagents (**1e** & **1f**). The iodinated compounds give much better yields (GC yields = 66% & 90%, respectively, UV, 8 h) compared to the brominated compounds, which agrees well with our observations in Main Text Table 1. When visible light (> 420 nm) is used as the photoexcitation source, both compounds require a reaction period of 12 h to give the yields of 76% and 74% of the corresponding deuterated compounds. We have included these results in Main Text Table 1 and added a short comment in discussion:

Page 8, Line 20: *“Both the electron-rich and -deficient iodinated substrates can be deuterated by D₂O (Table 1, 2a-2n).”*

Figure 1. Deuteration of brominated and iodinated silicon reagents.

Question 5: *What is the reactivity of bromothioanisole or iodothioanisole under this reaction conditions.*

Response: We have examined the reactivity of these two substrates for deuteration. *Iodothioanisole (1g)* can give excellent yield (GC yield > 99%) under UV irradiation for 8 h or visible light for 12 h. For *bromothioanisole*, we can only achieve medium yield (GC yield = 35%). The result of *Iodothioanisole (1g)* has been included in the revised Main Text Table 1:

Figure 2. Deuteration of bromothioanisole or iodothioanisole.

Question 6: *What is the reactivity ortho-dibromobenzene?*

Response: The reactivity of *ortho*-dibromobenzene is provided in Figure 3. We detect 61% of deuterated product after UV irradiation for 8 h.

Figure 3. Mono-deuteration of *ortho*-dibromobenzene.

Question 7: *Alkenyl halides could be deuterated?*

Response: We have tried to hydrogenate/deuterate alkenyl halides, such as (2-iodovinyl) benzene in our conditions in Figure 4, however, it doesn't work. We add a small amount of PdCl₂ as the co-catalyst for *in-situ* deposition of Pd nanoparticles on porous CdSe upon irradiation. We can observe a low amount hydrogenated product by using such catalyst, but the subsequent trial on deuteration is not successful.

Figure 4. Hydrodehalogenation of (2-iodovinyl)benzene.

Question 8: *The authors should explain why the reaction is chemoselective regarding different C–X bonds. Indeed, the optimized conditions allow the deuteration of aryl bromides, and aryl chloride but when 1-bromo-4-iodobenzene was used as substrate only C–I bond was touched.*

Response: Thank you for the comments on chemo-selectivity of different C–X bonds. In principle, C–I bond is much easier to cleave than C–Br, C–Cl and C–F bonds due to its lower bond dissociation energy (Supplementary Table 2). The iodinated compounds also have a more positive reduction potential than other halides (*Science* **2014**, 346, 725-728; *CRC handbook series in organic electrochemistry*. CRC press, **1977**), suggesting they can be reduced in a milder condition. Therefore, most iodinated substrates can be deuterated under the optimized conditions in Main Text Tables 1 & 2, while brominated substrates have remarkably lower reactivity in comparison to iodinated substrates; only those chlorinated and fluorinated substrates **with strong electron withdrawing groups** can be successfully deuterated.

For 1-bromo-4-iodobenzene (**Iaj**), we observe that only the C–I bond is preferentially reacted after **8 h**. Trace amount of the deuterodebrominated product can be detected at the 12th h (although not possible for yield calculation from the integration of GC-MS spectrum). For 1-chloro-4-iodobenzene (**Iai**) and 1-fluoro-4-iodobenzene (**In**), we cannot detect any deuterodechlorinated or deuterodefluorinated product even after reacting for 20 h. As mentioned above, we believe the reduction potentials of *chlorobenzene* or *fluorobenzene* are too negative for the photocatalytic deuteration. For *bromobenzene*, the reduction potential is quite close to the conduction band of porous CdSe so that trace amount of product can be detected. However if the substrate has **strong electron withdrawing groups, then aryl chlorides and aryl fluorides can also be deuterated**. A more detailed discussion on the chemo-selectivity regarding different C–X bonds has been included in the revised version:

Page 8, Line 9: *“This chemo-selectivity may be attributed to the different reduction potentials and bond dissociation energies of C–X bonds³⁷.”*

Question 9: *These results, which are accurately documented within the Supporting Information, are impressive and set a new benchmark for the specific deuteration, particularly within challenging aliphatic chains. Therefore, I recommend its publication in nature communication after answer to the above comments.*

Response: Thank you for the positive comments and advices. We believe we have addressed your questions in the revision.

Reviewer #3:

This communication reports an elegant approach for deuterium labelling through photoredox reactions catalysed by newly designed porous semiconducting (CdSe) nanosheet and using heavy water as deuteration agent to convert C-X to C-D efficiently and selectively, which shows distinct mechanism compared to traditional C-H/C-D exchange strategy. The reviewer believes that this method would significantly inspire new ideas of production of deuterated compounds in organic chemistry, which will thus impact both academic research and industrial manufacturing.

The data and methodology is adequate to support the conclusion. The figures and tables are organized well. However, the following minor comments need to be addressed to further improve the quality of this communication:

Response: We thank the reviewer for his/her positive comments. We have addressed his/her concerns on the control experiments using other sacrificing agents, mechanistic perspective on selectivity and reaction setup in this revised version. We have also modified the Supplementary Table 5 for better readability.

Question 1: *The effect of the sacrificing agent (additive) as electron donor on the catalytic performance should be discussed more in the text. Only two additives, Na₂SO₃ and OA were used in this study. What about other popular additives like triethylamine, etc?*

Response: Na₂SO₃ is the most commonly used sacrificial agent for cadmium chalcogenides in photocatalytic water splitting (*Inorg. Chem. Front.* **2016**, 3, 591-615) because it can donate an electron to the catalyst as well as protect the photocatalyst from degradation (*ACS Nano* **2013**, 7, 4316-4325). Another important reason for choosing Na₂SO₃ in this study is, anhydrous Na₂SO₃ does not contain any hydrogen atom. Such hydrogen atom can serve as potential hydrogen donor and significantly lower the deuteration efficiency (e.g., in the case of triethylamine).

To address the reviewer's concerns, we have supplied new control experiments using triethylamine (EtN₃), Na₂S·9H₂O, methanol (MeOH) and lactic acid in the revision, which are popular sacrificing agents in photocatalytic water splitting (*Chem. Rev.* **2010**, 110, 6503-6570; *Chem. Soc. Rev.* **2014**, 43, 7787-7812). For the sake of screening sacrificing agents, the experiments were conducted in UV light. Briefly, the use of triethylamine does not affect the hydrodehalogenation reaction and > 99% of yield and conversion can be observed. Both Na₂S·9H₂O and lactic acid give low yields of the hydrogenated products (~ 27%), while methanol is not working at all. This is reasonable because the oxidation potentials of triethylamine and Na₂SO₃ are 0.67 V (*C. R. Chimie* **2017**, 20, 283-295) and 0.389 V (*Environ. Health. Perspect.* **1985**, 64, 209-217) vs. SCE, respectively. They are much lower than the VB potential of porous CdSe nanosheets (1.1 V vs. SCE), which allows the donation of an electron to the photo-generated hole under light-excitation to recover its ground state for next catalytic cycle. Na₂S may poison the active Cd site due to the coordination. In the case of methanol and lactic acid, we can observe a significant catalyst degradation (color change from orange to black after short period).

Although triethylamine does not affect the hydrodehalogenation reaction, it can donate protons to the radical intermediates and result in much lower deuteration efficiency (49%). In this regard, we choose Na₂SO₃ as the sacrificing agent in our study.

Question 2: *It is surprising to see the super high selectivity in this transformation, such as arylamine, alkene and alkyne, which are generally reactive in photoredox catalysis. It is necessary to elaborate more form a mechanistic perspective, and give the specific reaction conditions if necessary.*

Response: Thank you so much for the critical comments on the selectivity. In follow we will explain the super high selectivity for our C-X to C-D transformations:

1) For arylamine, it is primarily a matter of **competition with the sacrificial agent** (Na₂SO₃). The VB potential of porous CdSe is ~ 1.1 V vs. SCE, which is apparently higher than the oxidation potential of Na₂SO₃ (0.389 V vs. SCE, *Environ. Health. Perspect.* **1985**, **64**, 209-217). Since arylamine has a very similar oxidation potential to CdSe (e.g., 0.9 V vs. SCE for aniline), the photo-generated holes from porous CdSe will preferentially transfer to the sacrificial agent (Na₂SO₃) rather than to the arylamine. Additionally, Na₂SO₃ is in large excess and is used in much higher concentration than arylamine, therefore we can observe a good functional group tolerance for arylamine.

2) For alkenes and alkynes, they usually **require noble metal catalysts for activation** to react with H or D radicals (*Chem. Commun.* **2016**, **52**, 1800-1803). Although the addition between alkenes/alkynes and aryl anion/aryl radicals is possible in bulk solution with non-noble catalysts (*J. Org. Chem.* **2014**, **79**, 9104-9111; *Tetrahedron Lett.* **2014**, **55**, 3355-3357; *Tetrahedron Lett.* **2013**, **54**, 2419-2422), such addition is **kinetically unfavorable** compared to the fast radical coupling between deuterium and carbon radicals. As stated in the Response to Reviewer 1, the radical coupling occurs on porous CdSe surface. The D radical (D₂O) generated *in-situ* on the surface of CdSe is in excess amount in practice. The reaction kinetics for water splitting is also faster than the cleavage of organic halides. So that the aryl anion/aryl radical, once generated, will quickly react with the nearby D radicals to form the deuterated compounds. There's no spare carbon radicals for alkenes or alkynes to form adducts in such case. Details on the surface-induced selectivity can be found in this review (*Acc. Chem. Res.* **2017**, **50**, 1002-1010).

To clarify the super high selectivity in our approach, we have included a short discussion in the revision:

Page 11, Line 3: "The excellent arylamine group tolerance (Table 1, 2h) can be attributed to the competition with the sacrificial agent (Na₂SO₃). Since Na₂SO₃ has an apparently lower oxidation potential than the valence band potential of CdSe and the oxidation potential of arylamine⁴³, the photo-generated holes will preferentially transfer to the sacrificial agent, which is in excess, rather than to the arylamine. For alkenes and alkynes (Table 2, 2am & 2an), the super high selectivity is due to the lack of noble metal catalysts for activation⁴⁴, as well as the kinetically unfavorable radical addition compared to the fast radical coupling."

Question 3: *Supplementary Table 5 is a little bit heavy. Actually some entries are not essential to display. For examples, Entry 1 using water only did not give any product because of solubility issue of the halo-compound, which has been mentioned in the text. Entries 3-5 are longer time reaction results for Entry 6. There is no difficulty to understand but easy to confuse the readers.*

Response: We agree that the original version is confusing to the readers and have removed some inessential entries in the revised version. Please see the revised version in the Supplementary Information or below:

Supplementary Table 5. Control experiments on photocatalytic hydrodehalogenation reaction ^{a)}

Entry	Catalyst	Loading	Additive	Solvent	Time	Conversion	GC Yield
1	Porous 2D CdSe	5 mg	Na ₂ SO ₃	CH ₃ CN/H ₂ O	2 h	> 99%	> 99%
2	Porous 2D CdSe	5 mg	Na ₂ SO ₃	CH ₃ CN/H ₂ O	1 h	64%	64%
3	CdSe nanoparticles	5 mg	Na ₂ SO ₃	CH ₃ CN/H ₂ O	2 h	36%	36%
4 ^{b)}	CdSe nanoplates	5 mg	Na ₂ SO ₃	CH ₃ CN/H ₂ O	2 h	trace	trace
5 ^{c)}	CdS powders	5 mg	Na ₂ SO ₃	CH ₃ CN/H ₂ O	2 h	28%	28%
6 ^{d)}	TiO ₂ nanopowders	5 mg	Na ₂ SO ₃	CH ₃ CN/H ₂ O	2 h	trace	trace
7 ^{e)}	Porous 2D CdSe	5 mg	Na ₂ SO ₃	CH ₃ CN/H ₂ O	2 h	trace	trace
8	--	--	Na ₂ SO ₃	CH ₃ CN/H ₂ O	2 h	trace	trace
9	Porous 2D CdSe	5 mg	--	CH ₃ CN/H ₂ O	2 h	27%	27%
10 ^{f)}	Porous 2D CdSe	5 mg	OA, OLA	CH ₃ CN/H ₂ O	2 h	40%	40%
11 ^{g)}	Porous 2D CdSe	5 mg	Na ₂ SO ₃	CH ₃ CN/H ₂ O	2 h	trace	trace
12 ^{h)}	Porous 2D CdSe	5 mg	Na ₂ SO ₃	CH ₃ CN/H ₂ O	8 h	> 99%	> 99%

^{a)} Standard conditions: **Ic** (0.1 mmol), CdSe nanosheets (5.0 mg), Na₂SO₃ (0.25 M), CH₃CN/H₂O (2.5 mL/2.5 mL), irradiation 2 h (> 280 nm), r.t.; ^{b)} CdSe nanoplates according to *Ref 5*; ^{c)} Commercial CdS powders from Sigma Aldrich (208183); ^{d)} Commercial TiO₂ nanopowders from Sigma Aldrich (718467); ^{e)} In the dark; ^{f)} 10.0 mmol OA and 10.0 mmol OLA instead of Na₂SO₃, OA = octylamine, OLA = oleylamine; ^{g)} In the dark under 1 balloon of H₂ for 24 h; ^{h)} > 420 nm.

Question 4: *The reaction setup is missing. Considering the experimental reproduction from others, it is important to provide the specification, as well as light intensity or supply power for UV light source.*

Response: Thank you so much for the concerns on reaction setup, which is very important for the reproducibility of this manuscript. We have included a more detailed description on the reaction conditions, specifications on light source, and digital photos of reaction setup in the Supplementary Information, see below:

**Supplementary Figure 19.** UV-Vis spectrum of the cut-off filter (> 420 nm) in this study.

Supplementary Figure 20. Digital photos of the reaction setup. (a, b) Setup for hydrodehalogenation and deuteration. The lamp is ~ 10 cm away from the container; (c, d) Setup for photocatalytic water splitting with automatically sampling to GC-MS. The Xenon lamp (150 W, $\sim 81 \pm 2 \text{ mW cm}^{-2}$ from light intensity meter) is water cooled to remove IR light. Selected cut-off filter can be installed between the lamp and reaction container.

Question 5: *Additionally, one recent review (chem Soc Rev 2016 (45), 6165-6212) comprehensively described the photoredox catalysis for organic and polymer synthesis. It is suggested to consider to cite this article.*

Response: We have already cited and commented on the above review, as follow:

Page 4, Line 2: *“A comprehensive review on the photoredox reactions used in organic and polymer synthesis is provided by Xu and Boyer et al²³.”*

NMR Spectra for the substrates requested by Reviewer 2:

The substrates were synthesized following the reported procedures (*J. Am. Chem. Soc.* **2003**, *125*, 6058-6059; *Angew. Chem. Int. Ed.* **2016**, *55*, 10406-10409; *Tetrahedron Lett.* **2011**, *52*, 1993-1995), which can also be seen in Supplementary Methods 17 ~ 19.

Figure 5. ^1H NMR of 1-bromo-4-(trimethylsilyl)benzene.

Figure 6. ^{13}C NMR of 1-bromo-4-(trimethylsilyl)benzene.

Figure 7. ^1H NMR of (4-bromophenyl)dimethylsilane.

Figure 8. ^{13}C NMR of (4-bromophenyl)dimethylsilane.

Figure 9. ^1H NMR of 1-iodo-4-(trimethylsilyl)benzene.

Figure 10. ^{13}C NMR of 1-iodo-4-(trimethylsilyl)benzene.

Figure 11. ^1H NMR of (4-iodophenyl)dimethylsilane.

Figure 12. ^{13}C NMR of (4-iodophenyl)dimethylsilane.

Figure 13. ^1H NMR of iodothioanisole.

Figure 14. ^{13}C NMR of iodothioanisole.

Figure 15. ^1H NMR of (2-iodovinyl)benzene.

Figure 16. ^{13}C NMR of (2-iodovinyl)benzene.

Reviewers' Comment:

Reviewer #2 (Remarks to the Author):

The manuscript has been significantly improved. This manuscript reports on effective transformation of C–X bond into C–D bond using photocatalytic D₂O splitting.

- (i) This method must not be compared with C–H bond to C–D bond exchange, which generally required directing group for the C–H bond cleavage and expensive metal complexes.
- (ii) This method did not require stoichiometric amount of metal base (BuLi) and displays a broad functional group tolerance.
- (iii) In the revised manuscript, the results are more clearly presented. However, the new substrates have not been included in main manuscript.
- (iv) Other sacrificial agents have been evaluated.
- (v) The experimental part has been improved.

Overall, all referee's concerns have been appropriately addressed. No doubt that the present work by Loh will inspire worldwide researchers to develop his concepts. All these justify publication in Nature communication.

Reviewer #3 (Remarks to the Author):

This reviewer has no more comments for the article.

Point-by-point responses to comments for manuscript NCOMMS-17-22821A

"Controllable Deuteration of Halogenated Compounds by Photocatalytic D₂O Splitting"

Reviewer #2:

The manuscript has been significantly improved. This manuscript reports on effective transformation of C–X bond into C–D bond using photocatalytic D₂O splitting.

1. This method must not be compared with C–H bond to C–D bond exchange, which generally required directing group for the C–H bond cleavage and expensive metal complexes.

2. This method did not require stoichiometric amount of metal base (BuLi) and displays a broad functional group tolerance.

3. In the revised manuscript, the results are more clearly presented. However, the new substrates have not been included in main manuscript.

4. Other sacrificial agents have been evaluated.

5. The experimental part has been improved.

We thank the Reviewer for his/her positive assessment. The new substrates have been included in Table1 (**2e**, **2f** and **2g**) in the main text.

Reviewer #3:

"This reviewer has no more comments for the article."

We thank the Reviewer for his/her positive assessment.